# Do SGLT2 Inhibitors Improve Cardiovascular Outcomes After Acute Coronary Syndrome Regardless of Diabetes? A Systematic Review and Meta-Analysis

**DOI:** 10.3390/medicina61101866

**Published:** 2025-10-17

**Authors:** Ioana Maria Suciu, Silvia Ana Luca, Simina Crișan, Alina-Ramona Cozlac, Svetlana Stoica, Constantin Tudor Luca, Bogdan Timar, Dan Gaita

**Affiliations:** 1Doctoral School, “Victor Babeș” University of Medicine and Pharmacy, Eftimie Murgu Sq. No. 2, 300041 Timișoara, Romania; silvia.luca@umft.ro; 2Institute of Cardiovascular Diseases Timisoara, 300310 Timișoara, Romania; simina.crisan@umft.ro (S.C.); alina-ramona.cozlac@umft.ro (A.-R.C.); tanaly@gmail.com (S.S.); ctluca@cardiologie.ro (C.T.L.); dgaita@cardiologie.ro (D.G.); 3Research Center of the Institute of Cardiovascular Diseases Timișoara, 300310 Timișoara, Romania; 4Cardiology Department, “Victor Babeș” University of Medicine and Pharmacy, Eftimie Murgu Sq. No. 2, 300041 Timișoara, Romania; 5Department of Diabetes, Nutrition and Metabolic Diseases Clinic, “Pius Brînzeu” Emergency Clinical County University Hospital, 300723 Timișoara, Romania; bogdan.timar@umft.ro; 6Department of Second Internal Medicine—Diabetes, Nutrition, Metabolic Diseases and Systemic Rheumatology, “Victor Babeș” University of Medicine and Pharmacy, Eftimie Murgu Sq. No. 2, 300041 Timișoara, Romania

**Keywords:** SGLT2 inhibitors, T2DM, acute coronary syndrome, all-cause mortality, CV mortality, recurrent MI

## Abstract

*Background and Objectives*: This systematic review and meta-analysis aims to evaluate whether the benefits of sodium–glucose co-transporter-2 (SGLT2) inhibitors on cardiovascular outcomes extend when initiated in patients with acute coronary syndrome (ACS), regardless of diabetic status. *Materials and Methods*: PubMed, Embase, and the Cochrane Library were searched from 2015 up to July 2025, according to PRISMA 2020 guidelines. Eligible studies were randomized controlled trials (RCTs) and observational studies comparing SGLT2 inhibitors with controls in post-ACS patients. Articles without full-text data for extraction, with unavailable outcome data or evaluating patients with stable coronary artery disease (CAD) were excluded. Primary outcomes were all-cause and cardiovascular (CV) mortality. Secondary outcomes included recurrent myocardial infarction (MI), rehospitalization for ACS, revascularization and stroke. Meta-analysis was conducted using the R statistical software (Version 4.5.1). Subgroup analysis was performed by study design to evaluate outcomes in type 2 diabetes mellitus (T2DM) populations. Risk of bias was assessed using the Cochrane Risk of Bias (RoB) 2.0 and Risk of Bias In Non-randomized Studies of Interventions (ROBINS-I) tools. Certainty of evidence was evaluated using the Grading of Recommendations, Assessment, Development and Evaluation (GRADE) approach. *Results*: A total of 16 studies were included in the meta-analysis, encompassing over 130,000 patients. Initiation of SGLT2 inhibitors after ACS was associated with a significant reduction in the primary outcome of all-cause mortality [hazard ratio (HR) = 0.77; (95% confidence interval (CI): 0.67–0.89)] and CV mortality [HR = 0.83; (95% CI: 0.70–0.99)]. In subgroup analyses, patients with T2DM experienced a significant reduction in all-cause mortality [HR = 0.73, (95% CI: 0.62–0.86)] and recurrent MI [HR = 0.83, (95% CI: 0.69–0.99)]. *Conclusions*: Initiation of SGLT2 inhibitors after ACS is associated with a significant reduction in all-cause and CV mortality. Subgroup analysis further demonstrated a reduction in all-cause mortality and recurrent myocardial infarction among patients with T2DM, while in patients without diabetes, no significant effects were observed. Although evidence certainty ranged from low to moderate and large RCTs are still ongoing, these findings support the early introduction of SGLT2 inhibitors in eligible patients with T2DM following ACS, pending confirmation by large, prospective clinical trials.

## 1. Introduction

Sodium–glucose co-transporter-2 (SGLT2) inhibitors have demonstrated significant benefits beyond glucose-lowering effects and are widely endorsed in clinical guidelines for patients with heart failure (HF), chronic kidney disease (CKD) and type 2 diabetes mellitus (T2DM), demonstrating reductions in the risk of hospitalization for heart failure (HF) and cardiovascular (CV) death, particularly in patients with established atherosclerotic cardiovascular disease [1,2,3,4,5,6,7,8].

Acute coronary syndrome (ACS), encompassing unstable angina (UA), ST-elevation myocardial infarction (STEMI) and non-ST-elevation myocardial infarction (NSTEMI), remains a major cause of morbidity and mortality worldwide. Despite advances in early revascularization and secondary prevention strategies, residual cardiovascular risk persists. Emerging evidence indicates that early initiation of SGLT2 inhibitors after acute myocardial infarction (AMI) may confer additional cardiovascular benefits, including reduced risk of CV death, all-cause mortality and recurrent ischemic events [9]. Although several observational studies have suggested improved outcomes with SGLT2 inhibitors after AMI, data from randomized controlled trials (RCTs) remain limited and inconclusive [10,11,12,13]. Furthermore, the impact of these agents in patients without diabetes who experience ACS is not well established.

While revascularization is not a primary endpoint in most SGLT2 inhibitors trials, several recent studies have reported data on this outcome [14]. The potential impact of SGLT2 inhibitors on atherosclerosis progression and the need for coronary revascularization procedures (percutaneous coronary intervention or coronary artery bypass graft) may be explained by their several pleiotropic effects that may contribute to reduced atherosclerotic burden and ischemic events.

Improvement in the endothelial function and myocardial energetics, reduction in oxidative stress, reductions in systolic blood pressure and body weight, decreased arterial stiffness and inflammation, favorable effects on lipid profile and uric acid levels, are all mechanisms which suggest a potential role in slowing coronary artery disease progression which could reduce the need for future revascularization [15,16]. This systematic review and meta-analysis aim to evaluate the effect of SGLT2 inhibitors on cardiovascular outcomes after ACS, regardless of diabetic status, by synthesizing evidence from both real-world observational data and RCTs.

## 2. Materials and Methods

### 2.1. Eligibility Criteria

Inclusion criteria: We included RCTs and observational studies with a follow-up ≥6 months. The population of interest was represented by patients (≥18 years) admitted for ACS, including (UA, STEMI or NSTEMI). Eligible studies evaluated treatment with SGLT2 inhibitors versus control (placebo or standard medical treatment) after the index ACS event and were included if they reported at least one of the outcomes. The primary outcomes were all-cause mortality and CV mortality following ACS in patients treated with SGLT2 inhibitors (death from any cause and death due to cardiovascular causes). Secondary outcomes included: recurrent myocardial infarction (non-fatal MI after the index hospitalization), rehospitalization for ACS, revascularization procedures and stroke incidence. For synthesis, studies were grouped according to study design (RCT vs. observational), population characteristics (T2DM, mixed cohorts or no T2DM) and timing of SGLT2 inhibitor initiation after the index ACS event (during hospitalization vs. early post-discharge). Subgroup analyses were performed by study design (RCT vs. observational) and based on diabetes status (T2DM vs. non-diabetic). In studies with mixed population, we included only the ACS subgroup is stratified results were available.

The timing of SGLT2 inhibitor initiation varied across included studies. Most initiated therapy during the index hospitalization, typically within 3–7 days after the ACS event, while others started treatment during the early post-discharge period, up to 4 weeks after the index event. We extracted and reported these initiation windows where available to account for potential differences in treatment effect related to timing.

Exclusion criteria: We excluded studies enrolling patients without a documented ACS event, studies lacking comparator group, articles with no publicly available full-text data for extraction, meta-analyses, case-reports, studies with unavailable outcome data, animal studies and studies evaluating SGLT2 inhibitors on patients with stable coronary artery disease (CAD), as well as, those in which outcomes could not be extracted separately. If subgroup-specific outcomes were not reported, such studies were excluded from the meta-analysis.

### 2.2. Search Strategy and Data Sources

This systematic review and meta-analysis was conducted in accordance with the Preferred Reporting Items for Systematic Reviews and Meta-Analyses (PRISMA) 2020 guidelines and the study selection process is summarized in the PRISMA flow diagram (see Figure 1). The protocol was not registered in PROSPERO, as the review process had already been initiated before registration. A comprehensive literature search was performed from 2015 until July 2025, with PubMed used as the primary database for study identification. Additional databases (Embase, Cochrane Library, and Web of Science) as well as references of relevant articles were also screened to identify additional eligible studies.

We aimed to identify studies evaluating the effects of SGLT2 inhibitors in patients with ACS, including both patients with and without diabetes. Search terms included (“dapagliflozin” or “Forxiga” or “empagliflozin” or “Jardiance” or “canagliflozin” or “SGLT2-I” or “SGLT 2 I” or “SGLT 2 inhibitors” or “Sodium Glucose Transporter 2 inhibitors” or “Sodium Glucose Transporter”) and (“Coronary Syndrome” or “Myocardial Infarction” or “Acute Coronary Syndrome” or “MI” or “ACS” or “Myocardial Ischemia” or “Ischemic Heart Disease” or “ST Elevation Myocardial Infarction” or “ST Segment Elevation Myocardial Infarction” or “ST Elevated Myocardial Infarction” or “STEMI” or “Non ST Elevation Myocardial Infarction” or “Non-ST-Elevation Myocardial Infarction” or “Non ST Elevated Myocardial Infarction” or “Non ST Elevated Myocardial Infarction” or “NSTEMI” or “Unstable Angina” or “unstable angina pectoris”) and (“mortality” or “all-cause mortality” or “cardiovascular mortality” or “CV mortality” or “CV death” or “revascularization” or “recurrent MI” or “rehospitalization” or “stroke”). This study was conducted following the principles outlined in the Declaration of Helsinki. Ethical approval was not required as it is a systematic review and meta-analysis of previously published data.

### 2.3. Selection Process and Data Collection

Two independent reviewers (I.M.S. and S.A.L.) screened titles and abstracts using Rayyan software, a web-based software for systematic reviews, followed by full-text review of potentially eligible studies [17]. Discrepancies were resolved by consensus. References were also screened to look for additional studies. After duplicate removal, the full texts were independently evaluated by the same reviewers against the inclusion and exclusion criteria and no automation tools were used. We extracted the following data in Microsoft Excel: study characteristics (author, year, country, design, sample size, follow-up duration), patient characteristics (age, gender, diabetes status, ACS subtype), intervention details (type and dose of SGLT2 inhibitors, timing of initiation) and outcomes (number of events and HRs with 95% CIs for each outcome of interest). Where multiple effect measures and multiple models were reported, we extracted HRs and their adjusted estimates. When outcomes were reported at different follow-up points, we used the longest follow-up. Because several studies did not reported data on SLGT2 inhibitors type and ACS subtype, we did not performed subgroup analyzes.

### 2.4. Reporting Bias Assessment and Certainty of Evidence

For RCTs, the risk of bias was assessed using the Cochrane Risk of Bias tool (RoB) version 2.0 tool, which contains five domains (randomization process, deviations from intended interventions, measurement of the outcome, missing outcome data and selection of the reported result). All five RCTs included demonstrated a low risk of bias (Figure 2). For observational studies we used the Risk of Bias In Non-randomized Studies of Interventions tool (ROBINS-I), which evaluates bias across seven domains (confounding, selection of participants, classification of intervention, deviations from intended interventions, missing data, measurement of outcomes and selection of the reported result) (Table 1). Overall, the risk of bias was moderate with the main source of concern being bias due to confounding, which will be considered when interpreting effect estimates. Each study was independently and manually evaluated by two reviewers (S.A.L., A.R.C.) and any disagreement was resolved by consensus. The overall risk of bias was categorized as low, moderate or high. Publication bias was evaluated both globally and within subgroups where there were more than ten studies, as Egger’s regression test and funnel plots are not reliable when fewer than ten studies are included.

We used the Grading of Recommendations, Assessment, Development and Evaluation (GRADE) approach to evaluate the certainty of the evidence across studies for the primary outcomes, with the following domains (risk of bias, inconsistency, indirectness, imprecision and publication bias), categorizing it as high, moderate or low.

### 2.5. Statistical Analysis

The statistical analysis was conducted using R statistical software (Version 4.5.1, R Foundation of Statistical Computing, Vienna, Austria). Overall, 16 studies were included in the meta-analysis assessing the benefits of SGLT2 inhibitors in improving cardiovascular outcomes after ACS. The pooled effect estimates were displayed in forest plots and reported as hazard ratios (HRs) and their 95% confidence intervals (CI), a *p* statistic value < 0.05 was considered statistically significant. Where effect estimates derived from PSM (propensity score matching) or IPTW (inverse probability of treatment weighting), we extracted these estimates to reduce confounding If unadjusted and adjusted estimates were reported, we extracted the adjusted model. The meta-analysis was performed when at least three studies reported the outcome, using the generic inverse variance method on the logarithm of HRs (log [HR]), with random effects modeling. To account for the expected clinical and methodological heterogeneity arising from the inclusion of both RCTs and observational studies, we used a random-effects model with inverse variance weighting, which provides more conservative estimates by incorporating between-study variability. Between-study variance was assessed using the restricted maximum likelihood (REML) estimator, and Knapp-Hartung adjustment was applied to provide more conservative CIs. The heterogeneity was assessed using Higgins I^2^ statistic test and considered low (0–25%), moderate (25–50%), substantial (50–75%) and high (>75%). In cases of substantial heterogeneity, we performed subgroup analyses according to population (T2DM), study design (RCT vs. observational) and leave-one-out sensitivity analyses by excluding studies at high risk of bias. Due to limited number of studies, meta-regression was not performed.

## 3. Results

### 3.1. Study Search and Characteristics of Included Studies

The database search yielded 743 records, and 15 trials were identified through clinical trial registries. After removing duplicates and screening titles and abstracts, 475 full-text articles were assessed for eligibility and 23 studies were included in the systematic review. Finally, 16 studies met the inclusion criteria. A PRISMA 2020 flow diagram (Figure 1) illustrates the study selection process. The moment of SGLT2 initiation was defined as the time elapsed after the index ACS. Across studies, initiation occurred either during hospitalization or shortly after discharge, most often within the first 1–12 weeks after ACS.

The 16 included studies comprised 5 RCTs and 11 observational studies. The characteristics of the included studies are summarized in Table 2 and Table 3. Among the observational studies, the most frequent study designs were retrospective cohorts. The included studies were conducted across diverse geographic regions, including Europe, North America, and Asia, thereby reflecting diverse healthcare settings and patient populations. In total, more than 130,000 patients with ACS were analyzed, with individual study sample sizes ranging from 109 to 89,554 participants. The mean age of participants ranged from 58 to 69 years, and the majority of study populations consisted predominantly of male patients (approximately 65–80%). Regarding diabetes status, 11 studies included exclusively patients with T2DM [13,18,19,21,22,23,24,26,28,29,30], 1 study excluded patients with diabetes [10], and 4 studies enrolled mixed cohorts [11,20,25,27]. Most studies enrolled patients with STEMI, while NSTEMI patients were included in a smaller proportion. Two studies with zero events in both arms were excluded from the effect size estimation [31,32]. Risk of bias assessment showed that all RCTs were at low risk (31.25%), while among observational studies, 10 (62.5%) were rated as moderate risk, and 1 (6.25%) as low-to-moderate. However, most observational studies used statistical adjustment methods such as PSM (propensity score matching) or IPTW (inverse probability of treatment weighting), and reported adjusted HRs to minimize confounding. These differences highlight the current state of research and the complexity of the available evidence, underscoring the variability and limitations of existing data regarding the effects of SGLT2 inhibitors after ACS.

### 3.2. Primary Outcome

#### 3.2.1. All-Cause Mortality

In a time-to-event meta-analysis of 11 studies, SGLT2 inhibitors initiation after ACS was associated with a statistically significant 23% relative reduction in the hazard of all-cause mortality, [HR = 0.77 (95% CI: 0.67–0.89), *p* = 0.002], Figure 3A. Most individual studies favored SGLT2 inhibitors, with several achieving statistical significance. DAPA-MI trial [10] reported an HR > 1.0, but with a wide confidence interval crossing unity, likely reflecting imprecision due to methodological differences, as it focused on a lower-risk population with preserved ejection fraction and no diabetes, potentially explaining the divergent result.

Overall, the results are consistent and support the mortality benefit of SGLT2 inhibitors across a large real-world population. While between-study heterogeneity was substantial (I^2^ = 78.7%) suggesting variation in study design (RCT vs. observational), sample size, differences in population characteristics (age, comorbidities, and diabetes), variability in follow-up duration (ranging from 6 months to 3 years), and disparities in baseline event rates, the overall direction of effect consistently favored SGLT2 inhibitors. The 95% prediction interval remained favorable, suggesting that while the average effect is protective, some future studies might show null results depending on clinical context, such as ACS severity, type, background therapies (revascularization, guideline-directed medical therapy), or differences in patient comorbidities and demographics.

##### All-Cause Mortality—Subgroup Analysis by Diabetes Status

A predefined subgroup analysis assessed whether the benefit varied according to baseline diabetes status. In patients with diabetes, the subgroup analysis showed a pooled HR of 0.73 (95% CI: 0.62–0.86), *p* = 0.03, indicating a 27% relative reduction in the hazard of all-cause mortality, Figure 3B. Heterogeneity was low to moderate (I^2^ = 37%). Together, these findings reinforce a strong mortality benefit of SGLT2 inhibitors in patients with diabetes, consistent with prior cardiovascular outcome trials and real-world data. These results are also aligned with current guideline recommendations for SGLT2 inhibitors in patients with T2DM at high cardiovascular risk and are supported by large landmark RCTs such as EMPA-REG OUTCOME [3] and DECLARE-TIMI 58 [5].

In contrast, among patients without diabetes, the effect was not statistically significant. The pooled HR was 0.87 [0.58–1.30] (*p* = 0.34), with high heterogeneity (I^2^ > 84%) and wide prediction intervals, Figure 3B. Although the direction of effect remained consistent with a potential benefit, the absence of statistical significance and high variability reduce confidence in the observed effect in patients without diabetes.

#### 3.2.2. Cardiovascular Mortality

10 studies were analyzed for CV mortality outcomes, encompassing over 60,000 participants. Pooled analysis using a random-effects model showed a significant reduction in CV mortality among patients treated with SGLT2 inhibitors compared to controls (HR = 0.83; 95% CI: 0.70–0.99; I^2^ = 55.5%; *p* = 0.03), Figure 4A. These findings were consistent across studies and demonstrated low to moderate heterogeneity, supporting the robustness of the association between SGLT2 inhibitor therapy and improved cardiovascular survival.

##### CV Mortality—Subgroup Analysis by Diabetes Status

When stratified by diabetes status, convergent patterns emerged regarding the CV mortality benefit of SGLT2 inhibitors. Among patients with diabetes, the complementary analysis showed a pooled HR of 0.69 (95% CI: 0.45–1.06, *p* = 0.07). Although this did not reach statistical significance, and heterogeneity was moderate (I^2^ = 67.6%), Figure 4C. Despite the lack of statistical significance in the HR model, the evidence shows that SGLT2 inhibitors may confer CV mortality benefit in this high-risk population. In the subgroup without diabetes, the pooled HR was 0.88 [95% CI 0.73–1.07, *p* = 0.12], not reaching statistical significance, Figure 4B. However, the consistency in direction of effect favoring SGLT2 inhibitors in both analyses suggests a potential benefit even in patients without diabetes.

#### 3.2.3. Subgroup Analysis by Study Design

Heterogeneity was substantial in the analysis, suggesting variation in study design. Therefore, we performed subgroup analysis which showed similar results with less pronounced heterogeneity, Figure 5. Observational studies reported a more pronounced mortality benefit. The pooled HRs for all-cause mortality and CV mortality were 0.69 (95% CI: 0.58–0.83) and 0.71 (95% CI: 0.42–1.21). These findings suggest a potentially greater observed effect in real-world settings, possibly reflecting broader patient inclusion criteria, higher baseline cardiovascular risk, longer follow-up durations, or differences in adherence and concomitant therapies compared to the more controlled conditions of randomized trials. However, observational studies are inherently subject to residual confounding and lack of randomization.

In contrast, RCTs showed a non-significant trend toward benefit, with HR = 0.87 (95% CI: 0.69–1.11) and 0.87 (95% CI: 0.65–1.17). The limited number of RCTs (n = 5) reduces statistical power and limits firm conclusions. Nevertheless, the consistency in effect direction between RCTs and observational data supports the plausibility of a true benefit. Further large-scale, well-powered randomized trials in ACS populations are warranted to confirm these findings and to better define the role of SGLT2 inhibitors in this high-risk setting. Ongoing trials such as DAPA-MI [10] and EMPACT-MI [11] are expected to provide additional insights into the efficacy of SGLT2 inhibitors in post-ACS populations.

#### 3.2.4. Publication Bias Analysis

For all-cause mortality, Egger’s regression test was performed to assess publication bias and showed possible small-study effects (*t* = 2.29, *p* = 0.047). Although this *p*-value is marginally below the conventional threshold (*p* < 0.05), the result does not strongly support the presence of funnel plot asymmetry. To further evaluate the robustness of our findings, we visually inspected the funnel plots, assessed the certainty of evidence using the Grades of Recommendation, Assessment, Development and Evaluation (GRADE) evidence profile (Table 4), and conducted a leave-one-out sensitivity analysis. These approaches highlight the need for cautious interpretation of the observed mortality benefit, as it may be influenced by selective reporting. To quantify the potential impact, we applied the trim-and-fill method. The adjusted pooled HR for all-cause mortality was HR = 0.69 (95% CI: 0.58–0.83), slightly lower than the original estimate HR = (0.77 [0.67–0.89]). This finding indicates that publication bias may have led to a modest overestimation of the effect size. However, the direction of the effect remained unchanged, supporting the robustness of the observed association between SGLT2 inhibitor therapy and reduced all-cause mortality. In contrast, for CV mortality, Egger’s regression test showed no significant small-study effects (*p* = 0.66). The corresponding funnel plots are presented in Figure 6. Future large-scale RCTs and the publication of negative or neutral results are essential to provide a more balanced and accurate estimate of the effect of SGLT2 inhibitors on all-cause mortality after ACS.

#### 3.2.5. Leave-One-Out Sensitivity Analysis

A leave-one-out sensitivity analysis was performed to assess the robustness of the pooled HRs, Figure 7. Each of the studies was removed one at a time, and the meta-analysis was recalculated to determine the impact of individual studies on the overall estimate. The pooled HRs ranged from 0.75 to 0.80 and 0.81 to 0.85. For all-cause mortality, all confidence intervals remained consistent with a beneficial effect of SGLT2 inhibitors. The heterogeneity (I^2^) varied between 57.4% and 80.8%, indicating no substantial change in between-study variability. Notably, the exclusion of Mee et al. [27] led to the largest drop in heterogeneity (I^2^ = 57.4%), suggesting that this study may contribute more to overall heterogeneity, possibly due to its large weight (14.6%) and narrow confidence interval. However, its exclusion did not change the direction or significance of the overall effect (HR = 0.80; 95% CI: 0.69–0.93). For CV mortality, the heterogeneity (I^2^) varied between 44.7% and 58.7%. These findings confirm that no single study disproportionately influenced the results of the meta-analysis, supporting the robustness and reliability of the observed mortality benefit associated with SGLT2 inhibitors in ACS patients.

### 3.3. Secondary Outcomes

#### 3.3.1. Recurrent MI

The effect of SGLT2 inhibitors on the risk of recurrent MI after an ACS event was evaluated using HR-based meta-analysis, based on available data from 8 studies (n = 110,206). The pooled analysis demonstrated a not statistically significant reduction in recurrent MI associated with SGLT2 inhibitor use (HR = 0.71, 95% CI: 0.41–1.21, *p* = 0.16), corresponding to a 30% relative risk reduction, Figure 8A. Heterogeneity was high (I^2^ = 80.6%), and the prediction interval (0.22–2.29) suggests that most future studies would likely report a protective effect, although some may yield neutral results depending on population risk, follow-up duration, and treatment adherence.

##### Recurrent MI—Subgroup Analysis by Diabetes Status

Subgroup analysis in patients with T2DM revealed a significant benefit. The pooled HR was 0.83 (95% CI: 0.69–0.99, *p* = 0.04), with no observed heterogeneity (I^2^ = 0%) and a narrow prediction interval (0.65–1.04), supporting the generalizability of the effect across patients with diabetes, Figure 8B. These findings suggest that SGLT2 inhibitors may reduce the risk of recurrent MI in patients with T2DM following ACS, with greater consistency and certainty than in the overall population. Further dedicated studies focused on recurrent ischemic events as primary endpoints may help clarify this potential benefit.

In the subgroup without diabetes (n = 3), the pooled analysis did not reach statistical significance (HR = 0.45, 95% CI: 0.03–5.92, *p* = 0.31), Figure 8B. The effect direction remained favorable, suggesting a potential benefit of SGLT2 inhibitors in this population, albeit with high uncertainty. Due to the limited number of studies and wide confidence intervals, the evidence remains inconclusive in this subgroup.

#### 3.3.2. Rehospitalization for ACS

A total of 3 studies were analyzed regarding rehospitalization for ACS. All included patients had T2DM. The analysis did not demonstrate a statistically significant effect of SGLT2 inhibitors. The pooled HR was 0.65 [95% CI: 0.15–2.82, *p* = 0.33], with moderate heterogeneity (I^2^ = 65.1%) and a wide prediction interval (0.06–7.32), Figure 8C. Due to the limited sample size and wide variability in event rates, no definitive conclusions can be drawn regarding the impact of SGLT2 inhibitors on ACS-related rehospitalization in patients with T2DM. Further evidence from larger and more homogeneous cohorts is warranted.

#### 3.3.3. Revascularization Rates

5 studies reported data on revascularization procedures following acute coronary events in patients with T2DM. In the time-to-event meta-analysis, the use of SGLT2 inhibitors was associated with a non-significant reduction of 25% in the hazard of undergoing revascularization (HR = 0.75, 95% CI: 0.18–3.2, *p* = 0.61), Figure 8D. The analysis was characterized by substantial between-study heterogeneity (I^2^ = 77.3%), and the 95% prediction interval ([0.04–14.15]) indicated considerable uncertainty regarding the true effect size in future settings. Notably, one study by Kurozumi et al., (Kurozumi et al., 2023) reported a paradoxical increase in revascularization risk (HR = 6.78), contributing to the observed heterogeneity, but in the sensitivity analysis, after its exclusion, the results were similar [22]. Zhu et al. [25] reported a significant reduction in the need of revascularization in ACS patients treated with SGLT2 inhibitors, notably, only this study included both patients with and without diabetes, while the others were restricted to individuals with type 2 diabetes, Figure 8E. This could partially explain the stronger protective effect observed in Zhu’s cohort, potentially pointing to broader vascular benefits of SGLT2i beyond glycemic control, after its exclusion, the heterogeneity I^2^ = 43.5%. On the other hand, Kurozumi et al. reported significantly increased rates, but this study had small sample sizes and possible confounding. After excluding as well I^2^ = 0%.

#### 3.3.4. Stroke After ACS

The use of SGLT2 inhibitors was associated with a non-significant trend toward reduced stroke risk in patients with ACS. While the overall effect did not reach statistical significance, the point estimate consistently favored SGLT2 inhibitor use, and several individual studies demonstrated significant reductions in stroke risk. A time-to-event meta-analysis including 8 studies reporting HRs yielded a similar conclusion, with a pooled HR of 0.94 (95% CI: 0.83–1.05, *p* = 0.20), and low heterogeneity (I^2^ = 0%), Figure 8F This analysis supports the consistency of the direction of effect across methodologies, though neither approach demonstrated a statistically significant reduction in stroke risk. Further research is warranted to better define the cerebrovascular benefit of SGLT2 inhibitors in this population.

##### Stroke—Subgroup Analysis by Diabetes Status

In patients with T2DM, pooled analysis showed a trend toward reduced risk of stroke associated with SGLT2 inhibitors, although not statistically significant. Time-to-event meta-analysis including 6 studies also revealed a similar, non-significant effect (HR: 0.85 [95% CI: 0.71 to 1.02], *p* = 0.06; I^2^ = 0%), Figure 8G. These findings suggest a possible stroke risk reduction in patients with diabetes treated with SGLT2 inhibitors, although this not reach statistical significance.

In patients without diabetes, the pooled HR showed no statistically significant difference [HR 0.92 (95% CI: 0.1–8.46), I^2^ = 30.8%, *p* = 0.70], with a wide prediction interval reflecting between-study uncertainty, Figure 8G. These findings, although limited by the small number of studies and heterogeneity, do not support a stroke risk reduction among patients without diabetes.

## 4. Discussion

This meta-analysis, including both RCTs and observational studies, shows that, in the overall population, SGLT2 inhibitors initiation after ACS is associated with reduction in all-cause and CV mortality. In patients without diabetes, no statistically significant effects were observed for either outcome. In patients with T2DM, the reduction remained significant for all-cause mortality but not for CV mortality. Regarding recurrent MI, a benefit was identified among patients with T2DM, but not for the overall cohort. These findings suggest a potential role in post-ACS secondary prevention in patients with diabetes. Early initiation, particularly during the acute or subacute post-ACS phase, may mitigate ischemia–reperfusion injury and enhance myocardial recovery.

Several factors may explain these findings. First, the number of studies and participants without diabetes was considerably smaller, limiting statistical power to detect meaningful differences. Second, the absolute CV risk is generally lower in patients without diabetes, which may attenuate the magnitude of benefit from SGLT2 inhibition. Third, the pleiotropic cardioprotective effects of SGLT2 inhibitors (natriuresis, blood pressure reduction, and improved endothelial function) might require longer follow-ups durations or occur predominantly in patients with metabolic disturbances such as insulin resistance or hyperglycemia. Lastly, heterogeneity in baseline characteristics, timing of therapy initiation, and study design may have further diluted potential treatment effects in this subgroup.

### 4.1. Context of Other Evidence

Evidence from broader T2DM populations supports these results. Patients with diabetes have an increased CV mortality, with CV disease being the leading cause of death in this population. In the EMPA-REG OUTCOME trial, empagliflozin reduced CV mortality by 38% and all-cause mortality by 32% in patients with T2DM and established CVD [3]. Benefits persisted in subgroups with prior CABG and when stratified by risk categories using the Thrombolysis In Myocardial Infarction (TIMI) Risk Score for Heart Failure in Diabetes (TRS-HF_DM_), even if the absolute treatment effect was higher in the very-high risk category [33,34]. In extensive CAD, Chipayo-Gonzales et al. found reduced all-cause mortality (HR = 0.34) with SGLT2 inhibitors initiation at discharge and a trend toward lower CV mortality with no statistical significance [35].

Even if there are very few RCTs that have evaluated patients with ACS, emerging data on the impact of SGLT2 inhibitors following ACS have been recently published. Adel et al. conducted a double-blind, placebo controlled randomized trial on 93 T2DM patients with ACS. Over a follow-up of 6 months, Empagliflozin showed no significant reduction in all-cause mortality, coronary revascularization or rehospitalization for UA [14]. Chang et al. observed reduced rehospitalization for ACS and sudden cardiac death in a propensity-matched analysis [12]. Similar findings were reported in ischemic heart failure subgroups [36], acute HF following ACS [37], and registries such as SGLT2-I AMI PROTECT [38]. Cai et al. found lower all-cause mortality in AMI patients post-PCI (7.6% vs. 2.5%) [39].

Comparative data suggest advantages over DPP4 inhibitors [40], and combination therapy with GLP-1RA further reduces MACE and mortality [41,42]. Some evidence also points to a stronger impact on MI risk compared to GLP-1RA alone [43] and possible prevention of the no-reflow phenomenon post-PCI [44]. In the ATH-SGLT2i pilot, early dapagliflozin initiation improved endothelial function regardless of diabetes [15].

### 4.2. Limitations

Variability in study design, follow-up duration, and outcome definitions may have limited comparability across studies. The inclusion of observational studies introduced heterogeneity and potential bias, although several used PSM, IPTW, or multivariable adjustment in order to reduce confounding. Subgroup analyses by diabetes status were constrained by limited availability of stratified data, and drug-specific analyses could not be performed, so the results reflect a class effect rather than individual agents. Despite these limitations, the consistent direction of effect across studies supports the robustness of the overall findings.

The inclusion of RCTs and observational studies broadened the evidence base and enhanced external validity, but it also increased variability in patient selection, treatment strategies, and outcome definitions. This heterogeneity may have attenuated the precision of pooled estimates and should be considered when interpreting our findings. Moreover, several outcomes, including recurrent MI and stroke, were primarily driven by observational evidence, which may be subject to residual confounding and therefore should not be generalized to the entire ACS population without confirmation from randomized trials.

Due to the population included, subgroup analyses by diabetes status could not be performed for certain secondary outcomes, including rehospitalization for ACS and revascularization rates, as for these outcomes the population was represented only by patients with diabetes. As a result, the differential effect of SGLT2 inhibitors in these outcomes remains unclear.

Although trends toward benefit were observed in patients without diabetes, these results did not reach statistical significance and were characterized by substantial heterogeneity and wide confidence intervals. As such, any potential benefit in this subgroup should be interpreted with great caution and regarded as a hypothesis rather than a conclusion.

### 4.3. Implications for Future Research

Strict adherence to SGLT2 inhibitors reduces cardiovascular hospitalizations and mortality in T2DM patients with heart failure, ACS, or those undergoing revascularization or cardiac surgery [45]. Given the substantial residual cardiovascular risk in patients with T2DM, current findings support SGLT2 inhibitors as part of secondary prevention strategies in T2DM after ACS. However, evidence in patients without diabetes remains inconclusive. Ongoing and future RCTs will be essential to clarify their role in patients without diabetes, to define their place in post-ACS management algorithms and guide clinical decision-making.

## 5. Conclusions

SGLT2 inhibitors are associated with a reduction in all-cause and CV mortality, as well as recurrent MI, following ACS, particularly among patients with T2DM. The benefits in all-cause mortality and recurrent MI were consistent and clinically meaningful in patients with T2DM, and similar trends were observed in patients without diabetes, where the evidence remains limited and inconclusive, with pooled estimates not reaching statistical significance and characterized by substantial heterogeneity. No significant effects were observed for stroke, revascularization, or rehospitalization for ACS. Differences between RCTs and observational studies further highlight the influence of study design on effect estimates and underscore the need for cautious interpretation. These findings suggest that the observed mortality reduction may be largely driven by patients with T2DM, underscoring the potential role of glycemic status in modulating the therapeutic response to SGLT2 inhibition. Although SGLT2 inhibitors were associated with reduced CV mortality in the overall population, the results did not achieve statistical significance, indicating a possible class effect that predominantly manifests in patients with diabetes. While initiation of therapy post-ACS appears beneficial in patients with diabetes, current evidence does not support universal early use in patients without diabetes. Large, dedicated RCTs, especially in post-ACS patients without diabetes, are essential to clarify the risk–benefit profile, confirm these findings, and define the role of SGLT2 inhibitors in future post-ACS management strategies.

## Figures and Tables

**Figure 1 medicina-61-01866-f001:**
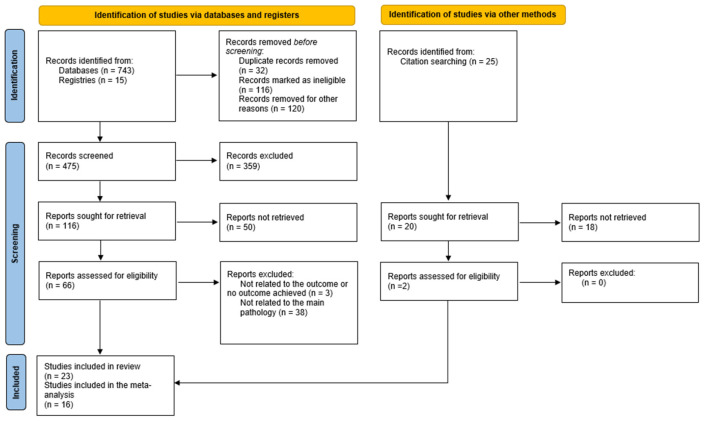
PRISMA 2020 flow diagram.

**Figure 2 medicina-61-01866-f002:**
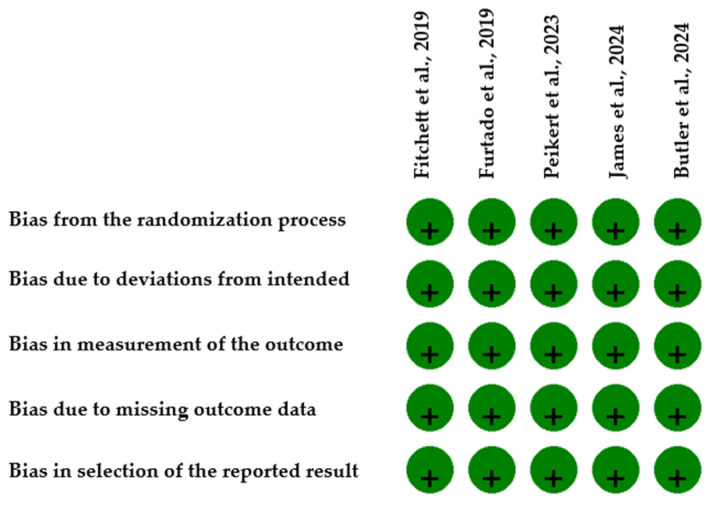
Risk of Bias for RCTs using RoB 2 [10,11,18,19,20]. “+” = Low risk of bias.

**Figure 3 medicina-61-01866-f003:**
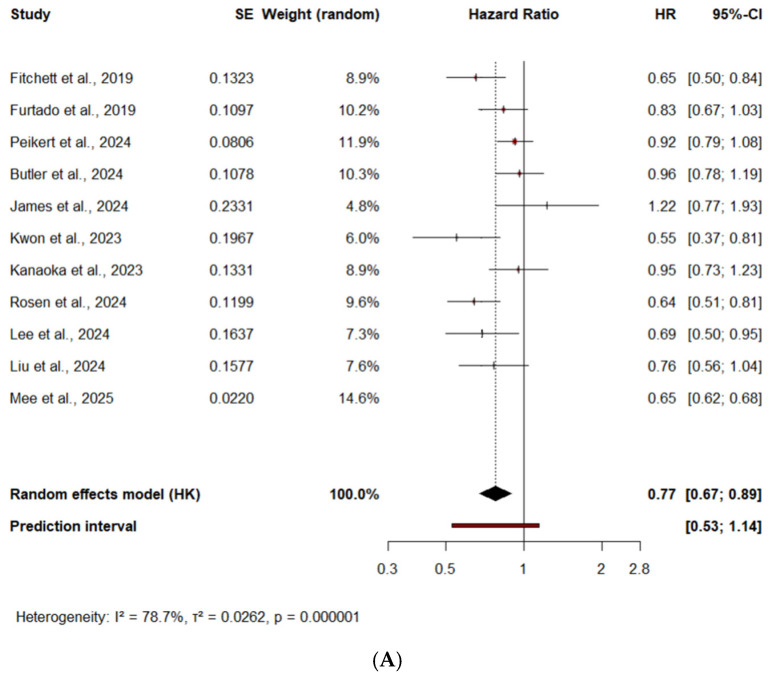
Forest Plot of all-cause mortality. (**A**) All-cause mortality overall. (**B**) All-cause mortality in T2DM patients and in patients without diabetes [10,11,18,19,20,21,24,27,28,29,30]. SE = Standard Error; HK = Hartung–Knapp adjustment; ■ = HR for each study, with the square size proportional to the study weight; — = 95% CI; ◆ = the pooled HR and its 95% CI for each subgroup and the overall analysis (random-effects model, Hartung–Knapp adjustment); vertical dashed line = line of no effect (HR = 1); I^2^ = percentage of variability due to heterogeneity; τ^2^ = between-study variance; *p* = significance of heterogeneity.

**Figure 4 medicina-61-01866-f004:**
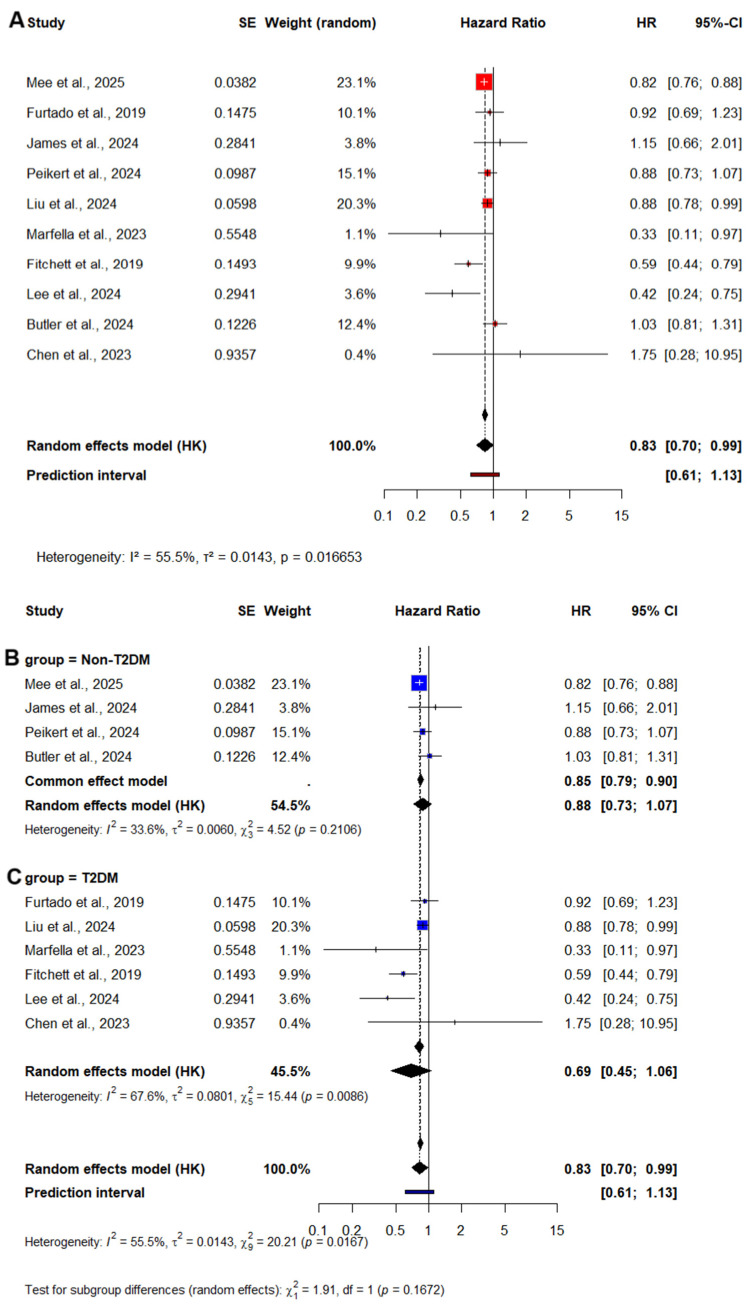
Forest Plot of CV mortality. (**A**) CV mortality overall. (**B**) CV mortality in patients without diabetes. (**C**) CV mortality in T2DM patients [10,11,13,18,19,20,26,27,28,29]. SE = Standard Error; HK = Hartung–Knapp adjustment; ■ = HR for each study, with the square size proportional to the study weight; — = 95% CI; ◆ = the pooled HR and its 95% CI for each subgroup and the overall analysis (random-effects model, Hartung–Knapp adjustment); vertical dashed line= line of no effect (HR = 1); I^2^ = percentage of variability due to heterogeneity; τ^2^ = between-study variance; *p* = significance of heterogeneity.

**Figure 5 medicina-61-01866-f005:**
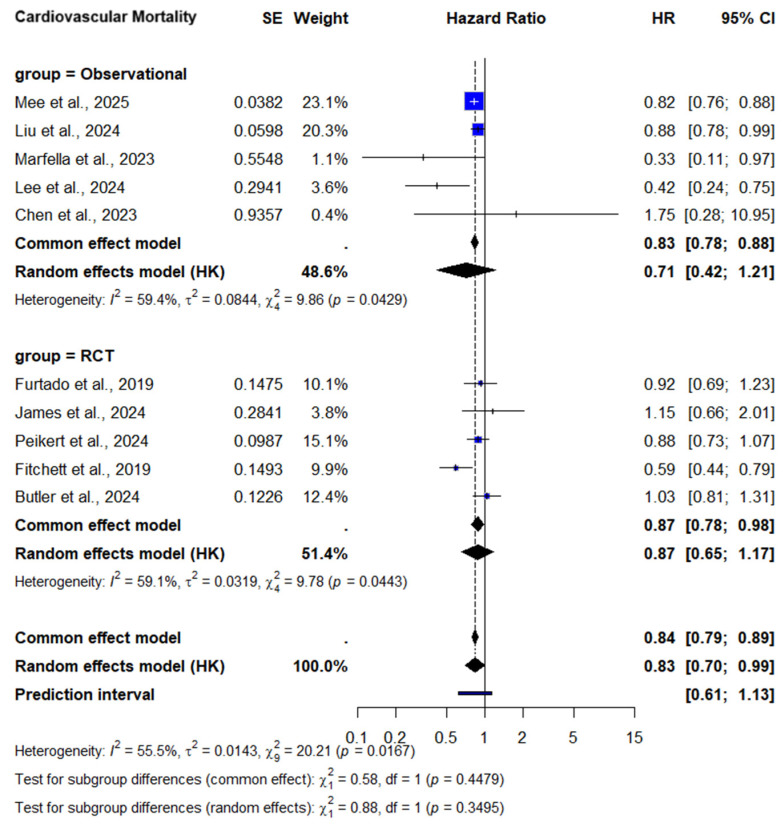
Subgroup Analysis by Study Design [10,11,18,19,20,21,24,27,28,29,30]. SE = Standard Error; HK = Hartung–Knapp adjustment; ■ = HR for each study, with the square size proportional to the study weight; — = 95% CI; ◆ = the pooled HR and its 95% CI for each subgroup and the overall analysis (random-effects model, Hartung–Knapp adjustment); vertical dashed line = line of no effect (HR = 1); I^2^ = percentage of variability due to heterogeneity; τ^2^ = between-study variance; *p* = significance of heterogeneity.

**Figure 6 medicina-61-01866-f006:**
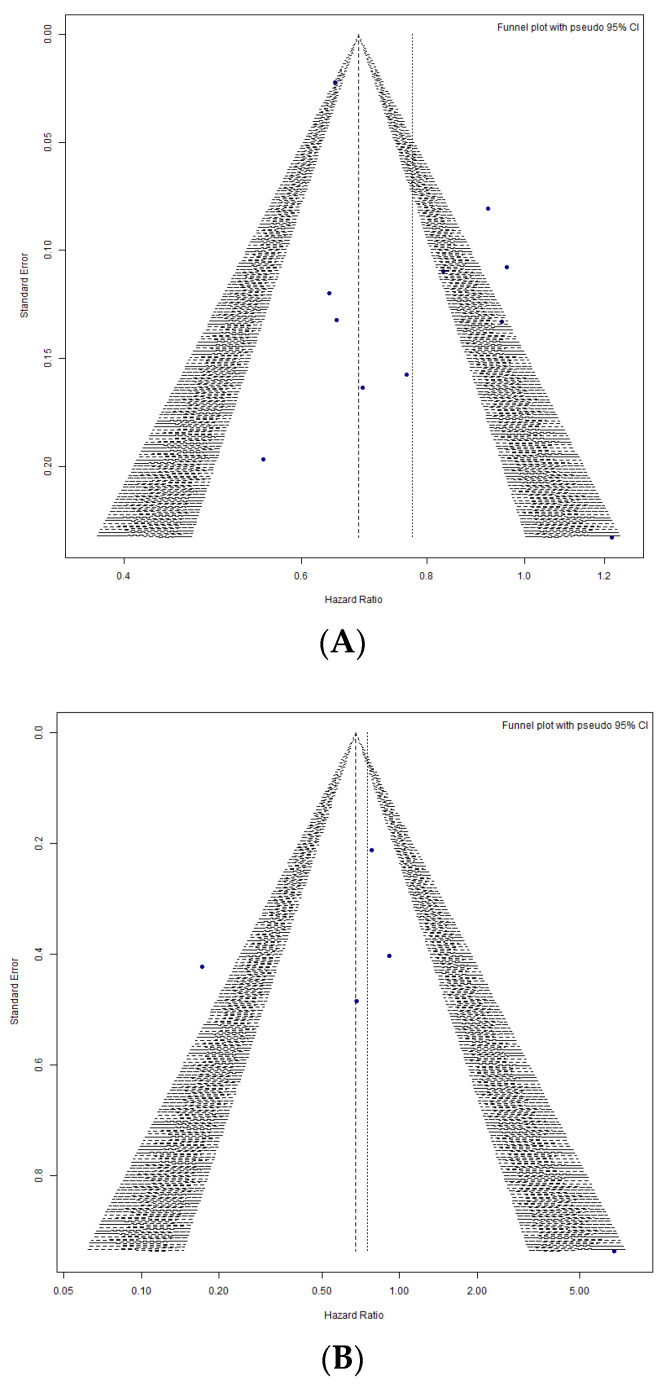
Funnel Plot. (**A**) All-cause Mortality. (**B**) CV-mortality. blue dots= individual study estimates, plotted according to their HR (x-axis) and SE (y-axis); solid vertical line= pooled HR; dashed diagonal lines= pseudo 95% confidence limits, within which study results are expected to fall in the absence of publication bias or small-study effects. Symmetry of the points around the central line suggests a low likelihood of publication bias, whereas asymmetry indicates potential selective reporting.

**Figure 7 medicina-61-01866-f007:**
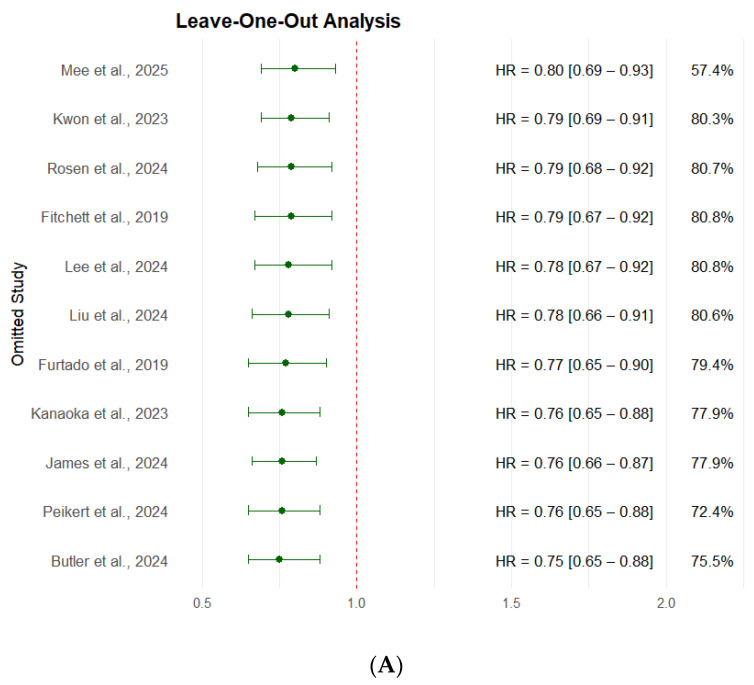
Leave-One-Out Sensitivity Analysis for the Primary Outcomes. (**A**) All-cause mortality. (**B**) CV mortality [10,11,13,18,19,20,21,24,26,27,28,29,30].

**Figure 8 medicina-61-01866-f008:**
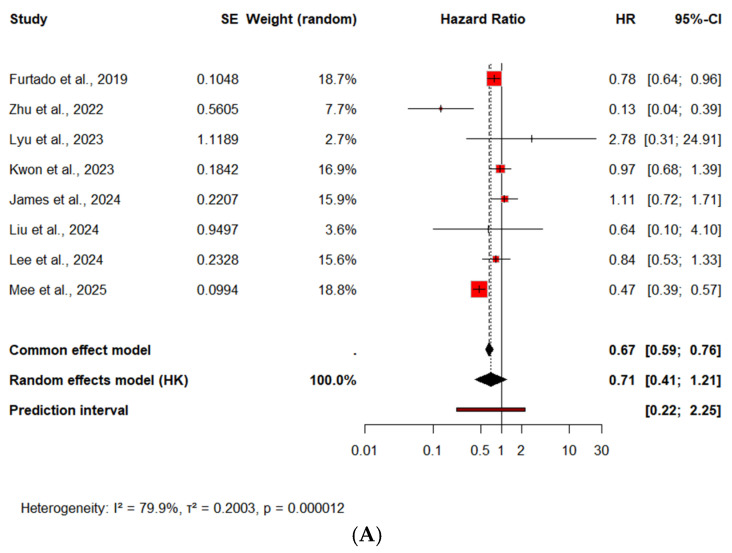
Forest Plot of the Secondary Outcomes. (**A**) Recurrent MI and (**B**) Subgroup analysis by diabetes status [10,19,23,25,27,28,29,30]. (**C**) Rehospitalization for ACS [13,21,26]. (**D**) Revascularization and (**E**) Subgroup analysis by diabetes status [22,23,25,28,29]. (**F**) Stroke and (**G**) Subgroup analysis by diabetes status [10,19,23,24,27,28,29,30]. SE = Standard Error; HK = Hartung–Knapp adjustment; ■ = HR for each study, with the square size proportional to the study weight; — = 95% CI; ◆ = the pooled HR and its 95% CI for each subgroup and the overall analysis (random-effects model, Hartung–Knapp adjustment); vertical dashed line = line of no effect (HR = 1); I^2^ = percentage of variability due to heterogeneity; τ^2^ = between-study variance; *p* = significance of heterogeneity.

**Table 1 medicina-61-01866-t001:** Risk of Bias for Observational Studies—ROBINS I.

Study	Bias Due to Confounding	Bias in Selection of Participants	Bias in Classification of Interventions	Bias Due to Deviations from Intended Interventions	Bias Due to Missing Data	Bias in Measurement of Outcomes	Bias in Selection of the Reported Result	Overall Bias
Chen et al. [13]	1.1 PY; 1.2 PY; 1.4 PY; 1.5 PN; 1.6 PN; 1.7 NA Moderate risk	2.1 Y; 2.2 Y; 2.4 Y Low risk	3.1 Y; 3.2 Y; 3.3 Y Low risk	4.1 PN; 4.3 PY; 4.4 PY; 4.5 Y Low risk	5.1 Y; 5.2 Y; 5.3 Y Low risk	6.1 N; 6.2 PY; 6.3 PY; 6.4 PN Low risk	7.1 PN; 7.2 PN; 7.3 PN Moderate risk	Moderate Risk
Kanaoka et al. [21]	1.1 PY; 1.2 PY; 1.4 PY; 1.5 PN; 1.6 PN; 1.7 NA Moderate risk	2.1 Y; 2.2 Y; 2.4 Y Low risk	3.1 Y; 3.2 Y; 3.3 Y Low risk	4.1 PN; 4.3 PY; 4.4 PY; 4.5 Y Low risk	5.1 Y; 5.2 Y; 5.3 Y Low risk	6.1 N; 6.2 PY; 6.3 PY; 6.4 PN Low risk	7.1 PN; 7.2 PN; 7.3 PN Moderate risk	Moderate Risk
Kurozumi et al. [22]	1.1 PY; 1.2 PY; 1.4 PY; 1.5 PN; 1.6 PN; 1.7 NA Moderate risk	2.1 Y; 2.2 Y; 2.4 Y Low risk	3.1 Y; 3.2 Y; 3.3 Y Low risk	4.1 PN; 4.3 PY; 4.4 PY; 4.5 Y Low risk	5.1 Y; 5.2 Y; 5.3 Y Low risk	6.1 N; 6.2 PY; 6.3 PY; 6.4 PN Low risk	7.1 PN; 7.2 PN; 7.3 PN Moderate risk	Moderate Risk
Lyu et al. [23]	1.1 PY; 1.2 PY; 1.4 PY; 1.5 Y; 1.6 Y; 1.7 NA Moderate risk	2.1 Y; 2.2 Y; 2.4 Y Low risk	3.1 Y; 3.2 Y; 3.3 Y Low risk	4.1 PN; 4.3 PY; 4.4 PY; 4.5 Y Low risk	5.1 Y; 5.2 Y; 5.3 Y Low risk	6.1 N; 6.2 PY; 6.3 PY; 6.4 PN Low risk	7.1 PN; 7.2 PN; 7.3 PN Moderate risk	Moderate Risk
Rosen et al. [24]	1.1 PY; 1.2 PY; 1.4 PY; 1.5 Y; 1.6 PN; 1.7 NA Moderate risk	2.1 Y; 2.2 Y; 2.4 Y Low risk	3.1 Y; 3.2 Y; 3.3 Y Low risk	4.1 PN; 4.3 PY; 4.4 PY; 4.5 Y Low risk	5.1 Y; 5.2 Y; 5.3 Y Low risk	6.1 N; 6.2 PY; 6.3 PY; 6.4 PN Low risk	7.1 PN; 7.2 PN; 7.3 PN Moderate risk	Moderate Risk
Zhu et al. [25]	1.1 PY; 1.2 PY; 1.4 PY; 1.5 Y; 1.6 PN; 1.7 NA Moderate risk	2.1 Y; 2.2 Y; 2.4 Y Low risk	3.1 Y; 3.2 Y; 3.3 Y Low risk	4.1 PN; 4.3 PY; 4.4 PY; 4.5 Y Low risk	5.1 Y; 5.2 Y; 5.3 Y Low risk	6.1 N; 6.2 PY; 6.3 PY; 6.4 PN Low risk	7.1 PN; 7.2 PN; 7.3 PN Moderate risk	Moderate Risk
Marfella et al. [26]	1.1 PY; 1.2 PY; 1.4 PY; 1.5 PN; 1.6 PN; 1.7 NA Moderate risk	2.1 Y; 2.2 Y; 2.4 Y Low risk	3.1 Y; 3.2 Y; 3.3 Y Low risk	4.1 PN; 4.3 PY; 4.4 PY; 4.5 Y Low risk	5.1 Y; 5.2 Y; 5.3 Y Low risk	6.1 N; 6.2 PY; 6.3 PY; 6.4 PN Low risk	7.1 PN; 7.2 PN; 7.3 PN Moderate risk	Moderate Risk
Mee et al. [27]	1.1 Y; 1.2 Y; 1.4 Y; 1.5 Y; 1.6 Y; 1.7 NA Low risk	2.1 Y; 2.2 Y; 2.4 Y Low risk	3.1 Y; 3.2 Y; 3.3 Y Low risk	4.1 PN; 4.3 PY; 4.4 PY; 4.5 Y Low risk	5.1 Y; 5.2 Y; 5.3 Y Low risk	6.1 N; 6.2 PY; 6.3 PY; 6.4 PN Low risk	7.1 PN; 7.2 PN; 7.3 PN Moderate risk	Low to Moderate Risk
Liu et al. [28]	1.1 Y; 1.2 PN; 1.4 Y; 1.5 PY; 1.6 PN; 1.7 NA Moderate risk	2.1 Y; 2.2 N; 2.4 Y Moderate risk	3.1 Y; 3.2 PN; 3.3 PN Moderate risk	4.1 PN; 4.3 PY; 4.4 PY; 4.5 Y Low risk	5.1 PY; 5.2 PN; 5.3 PN Low risk	6.1 N; 6.2 PY; 6.3 PY; 6.4 PN Low risk	7.1 PN; 7.2 PN; 7.3 PN Low risk	Moderate Risk
Lee et al. [29]	1.1 Y; 1.2 PN; 1.4 Y; 1.5 PY; 1.6 PN; 1.7 NA Moderate risk	2.1 Y; 2.2 N; 2.4 Y Moderate risk	3.1 Y; 3.2 PN; 3.3 PN Moderate risk	4.1 PN; 4.3 PY; 4.4 PY; 4.5 Y Low risk	5.1 PY; 5.2 PN; 5.3 PN Low risk	6.1 N; 6.2 PY; 6.3 PY; 6.4 PN Low risk	7.1 PN; 7.2 PN; 7.3 PN Low risk	Moderate Risk
Kwon et al. [30]	1.1 Y; 1.2 PN; 1.4 Y; 1.5 PY; 1.6 PN; 1.7 NA Moderate risk	2.1 Y; 2.2 N; 2.4 Y Moderate risk	3.1 Y; 3.2 PN; 3.3 PN Moderate risk	4.1 PN; 4.3 PY; 4.4 PY; 4.5 Y Low risk	5.1 PY; 5.2 PN; 5.3 PN Low risk	6.1 N; 6.2 PY; 6.3 PY; 6.4 PN Low risk	7.1 PN; 7.2 PN; 7.3 PN Low risk	Moderate Risk

**Table 2 medicina-61-01866-t002:** Summary of the Included Studies.

Author, Year & Country	Study Design and Type	Follow-Up (Months)	Group	Number	Age [Years ± SD; (IQR)]	Male (*n*. %)	T2DM (*n*. %)	Type of SGLT2 Inhibitor (*n*. %)	STEMI (*n*. %) NSTEMI (*n*. %)
Fitchett et al. [18] 2019 International (42 countries)	RCT	37.2	SGLT2	3048	62.8 ± 8.6	2189 (71.8%)	3048 (100%)	Empagliflozin 10 or 25 mg	NR
			Control	1518	63.0 ± 8.8	1098 (72.3%)	1518 (100%)	Placebo	NR
Furtado et al. [19] 2019 International (33 countries)	RCT	50.4	SGLT2	1777	62.0 (57.68)	1364 (76.75%)	1777 (100%)	Dapagliflozin 10 mg	NR
			Control	1807	62.0 (57.68)	1375 (76.09%)	1807 (100%)	Placebo	NR
Butler et al. [11] 2024 International (22 countries)	RCT EMPACT-MI	17.9	SGLT2	3260	63.6 ± 11.0	2448 (75.09%)	1035 (31.7%)	Empagliflozin 10 mg	2444 (75%) 814 (25%)
			Control	3262	63.7 ± 10.8	2449 (75.07%)	1046 (32.1%)	Placebo	2401 (73.6%) 861 (26.4%)
James et al. [10] 2024 Sweden & UK	RCT DAPA-MI	11.6	SGLT2	2019	63.0 ± 11.06	1631 (80.8%)	0%	Dapagliflozin 10 mg	1465 (72.6%) 544 (26.9%)
			Control	1998	62.8 ± 10.64	1579 (79%)	0%	Placebo	1428 (71.5%) 562 (28.1%)
Peikert et al. [20] 2024 USA	RCT	22.9	SGLT2	1830	68.7 ± 9.7	2825 (75.7%)	1835 (49.2%)	Dapagliflozin 10 mg	NR
			Control	1901				Placebo	NR
Mee et al. [27] 2025 USA	Observational Retrospective PSM 1:1	12	SGLT2	44,777	68.5 ± 11.9	30,699 (68.55%)	28,610 (63.9%)	NR	NR
			Control	44,777	68.8 ± 13.3	30,566 (68.26%)	29,407 (65.7%)	NR	NR
Zhu et al. [25] 2022 China	Observational Retrospective	23	SGLT2	141	60.6 ± 13.6	105 (74.5%)	96 (68.1%)	Dapagliflozin 10 mg	99 (70.2%) NR
			Control	645	62.5 ± 13.5	497 (77.1%)	96 (14.9%)	Never used	396 (61.4%) NR
Chen et al. [13] 2023 China	Observational Retrospective	10	SGLT2	128	64 (56.7)	96 (75%)	128 (100%)	Empagliflozin 10 mg 81 (63.3%) Dapagliflozin 10 mg 47 (36.7%)	51 (39.8%) 33 (25.8%)
			Control	104	67 (57.71)	79 (76.7%)	104 (100%)	Never used	54 (51.9%) 23 (22.1%)
Kanaoka et al. [21] 2023 Japan	Observational Retrospective PSM 1:1	24	SGLT2	1591	NR	1221 (76.74%)	1591 (100%)	NR	NR
			Control	1591	NR	1222 (76.8%)	1591 (100%)	Never used	NR
Kurozumi et al. [22] 2023 Japan	Observational	6	SGLT2	40	65.48 ± 13.46	32 (80%)	40 (100%)	Empagliflozin 10 mg 9 (22.5%) Dapagliflozin 10 mg 31 (77.5%)	29 (72.5%) 6 (15%)
			Control	69	73.81 ± 11.76	50 (72.46%)	69 (100%)	Never used	41 (59.4%) 17 (24.6%)
Kwon et al. [30] 2023 Korea	Observational PSM 1:2	25.2	SGLT2	938	56.4 ± 11.3	769 (82%)	938 (100%)	Empagliflozin 10 mg 302 (32.2%) Dapagliflozin 10 mg 605 (64.5%) Ipragliflozin 10 mg 31 (3.3%)	550 (58.6%) 388 (41.4%)
			Control	1876	57.6 ± 11.3	1482 (79%)	1876 (100%)	Non Users	1137 (60.6) 739 (39.4%)
Lyu et al. [23] 2023 Korea	Observational IPTW	12	SGLT2	186	59.11 ± 11.52	150 (80.7%)	186 (100%)	NR	100 (53.8%) NR
			Control	593	66.12 ± 10.86	422 (71.2%)	593 (100%)	Never used	227 (38.3%) NR
Marfella et al. [26] 2023 Italy	Observational Prospective	12	SGLT2	177	66.2 ± 6.3	115 (65%)	177 (100%)	NR	99 (55.9%) NR
			Control	200	65.4 ± 6.1	128 (64%)	200 (100%)	Never used	111 (55.5%) NR
Lee et al. [29] 2024 Taiwan	Observational PSM 1:1	from index date to the independent occurrence of the study outcomes, discontinuation of medication, or end of the study period	SGLT2	944	NR	NR	944 (100%)	Empagliflozin 10 mg (57%) Dapagliflozin 10 mg (38%) Canagliflozin 100 mg (5%)	NR
			Control	944	NR	NR	944 (100%)	Non Users	NR
Liu et al. [28] 2024 China	Observational, retrospective PSM 1:1	12	SGLT2	226	62.9 ± 10.8	143 (63.3%)	226 (100%)	NR	104 (46%) 68 (30.1%)
			Control	226	62.1 ± 11.7	129 (57.1%)	226 (100%)	Never used	110 (48.7%) 69 (30.5%)
Rosen et al. [24] 2024 Sweden	Observational registry study from SWEDEHEART	12	SGLT2	2498	69 (61–75)	1944 (77.8%)	2498 (100%)	Dapagliflozin 480 (19.2%) Empaglifozin 1951 (78.1%) Canagliflozin 8 (0.3%)	969 (38.8%) 1529 (61.2%)
			Control	8773	73 (65–79)	6084 (69.3%)	8773 (100%)	Never used	2671 (30.5%) 6101 (69.6%)

SD = standard deviation, IQR = interquartile range, PSM = Propensity Score Matching, IPTW = Inverse Probability of Treatment Weighting, NR = Not Reported.

**Table 3 medicina-61-01866-t003:** Summary of the Included Studies—Moment of SGLT2 Inhibitors Initiation.

Author & Year	Study Design and Type	Main Pathology	Moment of Initiation (Days or Weeks After ACS)
Fitchett et al. [18] 2019	Pre-specified sub-group analysis EMPA-REG OUTCOME	Previous MI	After 8 weeks
Furtado et al. [19] 2019	Prespecified sub-group analysis of DECLARE-TIMI 58 patients with previous MI and T2DM	Previous MI	After 8 weeks
Butler et al. [11] 2024	EMPACT-MI	AMI at risk for HF	Within 14 days
James et al. [10] 2024	DAPA-MI	AMI without T2DM or HF	Within 7–10 days
Peikert et al. [20] 2024	Planned participant-level pooled analysis of patients with previous MI and HF, DAPA-HF and DELIVER RCT sub-analysis	Previous MI	After 12 weeks
Mee et al. [27] 2025	Observational Retrospective PSM 1:1	AMI	Within the first 14 days after initial AMI
Zhu et al. [25] 2022	Observational Retrospective	AMI	At discharge
Chen et al. [13] 2023	Observational Retrospective	ACS	Before/during hospitalization
Kanaoka et al. [21] 2023	Observational Retrospective PSM 1:1	ACS	Within 2 weeks, used before 194 (12%)
Kurozumi et al. [22] 2023	Observational	ACS	During hospitalization
Kwon et al. [30] 2023	Observational PSM 1:2	AMI	Within 14 days AMI + T2DM treated with PCI
Lyu et al. [23] 2023	Observational IPTW	AMI	Discharge
Marfella et al. [26] 2023	Observational, Prospective	AMI	At least 6 months before
Lee et al. [29] 2024	Observational PSM 1:1	AMI	Within 12 weeks discharge
Liu et al. [28] 2024	Observational, retrospective PSM 1:1	ACS	During hospitalization
Rosen et al. [24] 2024	Observational registry study from SWEDEHEART	AMI	Within 120 days before hospital discharge, prescribed during hospitalisation or redeemed prescription within 3 days.

**Table 4 medicina-61-01866-t004:** GRADE Evidence Profile.

Outcome	No. of Studies	Certainty Rating Domains					HR [95% CI]	Overall Certainty of Evidence
		Risk Of Bias (RoB 2 and ROBINS I)	Inconsistency	Indirectness	Imprecision	Publication Bias		
All-cause mortality	5 RCTs + 6 observational (n = 130,288)	No concern	Serious ^a^	No concern	No concern	Serious ^b^	0.77 [0.67–0.89]	⬤⬤◯◯ Low
CV mortality	5 RCTs + 5 observational (n = 121,627)	No concern	Some concern ^c^	No concern	No concern	No concern	0.83 [0.70–0.99]	⬤⬤⬤◯ Moderate

^a^ Overall heterogeneity was high (I^2^ = 78.7%). ^b^ Publication bias was assessed using Egger’s test. ^c^ Overall heterogeneity was moderate (I^2^ = 55.5%). **Symbols:** ⬤ = filled circle; ◯ = empty circle. **GRADE certainty levels:** ⬤⬤⬤◯ moderate, ⬤⬤◯◯ low.

## Data Availability

No new data were created or analyzed in this study. Data sharing is not applicable to this article.

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
