# Peer review of "Do SGLT2 Inhibitors Improve Cardiovascular Outcomes After Acute Coronary Syndrome Regardless of Diabetes? A Systematic Review and Meta-Analysis"

_medicina, 2025, doi:10.3390/medicina61101866_

Round 1

Reviewer 1 Report

Comments and Suggestions for Authors

Thank you for the opportunity to review this systematic review about whether SGLT2 inhibitors improve cardiovascular outcomes regardless of diabetes status.

Below are my comments on the manuscript.

  1. In the Abstract, please provide the range of searching period (up to July 2025 since when?)
  2. Provide the references number in the sentences in linces 50-53.
  3. In Figure 1, there is incomplete information (databases n=?? and Registers=??).
  4. In Data screening and extraction authors explained that ACS subtype was extracted; however, what is the reason did not put the NSTEMI in Table 1 (Summary of the Included studies? Please provide the data of NSTEMI in Table 1.
  5. To improve the clarity, provide additional information in “moment of initiation” by mentioning, “days or weeks after ACS”.
  6. Provide the text about PRISMA search results in Results section, as Figure 1 has been not mentioned in the Methods not either in the Results.
  7. Add a brief explanation about characteristics of participants in Summary of included studies, including but not limited to design of study, the country of study, sample number in general, age range, and percentage of studies in each level of risk of bias.

Major comment

Because the aim of this study is to see the SGLT2 inhibitor effect after ACS events in patients with and without diabetes, it is critical to clearly state that in patients without diabetes, the pooled HRs of all-cause mortality, cardiovascular mortality, recurrent MI (despite its small number study observing recurrent MI as an outcome), and stroke  after SGLT2 inhibitor therapy were not statistically significant. Although authors have explained it in the Conclusions, but authors must properly state these findings and provide the possible reasons for why those insignificant results were found in participants without diabetes in Discussion as well. 

Author Response

Thank you for taking the time to review this manuscript. Please find the detailed responses below and the corresponding revisions highlighted in the re-submitted files.

Comments 1: In the Abstract, please provide the range of searching period (up to July 2025 since when?)

Response 1: Thank you for pointing this out. We clarified the search period in the abstract as follows: “were searched from 2015 up to July 2025, according to PRISMA 2020 guidelines”. (Abstract, line 25 and 2.2 Search Strategy and Data Sources, lines 117-118).

Comments 2: Provide the references number in the sentences in lines 50-53.

Response 2: We have, accordingly, revised the references and added the corresponding numbers. (Introduction, line 59).

Comments 3: In Figure 1, there is incomplete information (databases n=?? and Registers=??).

Response 3: We have now completed the figure with PRISMA flow diagram by providing the total number of records identified from databases and registers. (3.1. Study Search and Characteristics of Included Studies, updated Figure 2, page 8).

Comments 4: In Data screening and extraction authors explained that ACS subtype was extracted; however, what is the reason did not put the NSTEMI in Table 1 (Summary of the Included studies? Please provide the data of NSTEMI in Table 1.

Response 4: Thank you for pointing this out. We have now revised the table to include data on ACS subtypes, including the number or proportion of NSTEMI patients reported in each study (when available). (3.1. Study Search and Characteristics of Included Studies, updated Table 2, page 8).

Comments 5: To improve the clarity, provide additional information in “moment of initiation” by mentioning, “days or weeks after ACS”.

Response 5: We have clarified the “moment of initiation” column in updated Table 3 (3.1. Study Search and Characteristics of Included Studies, page 11), to specify the time frame of SGLT2 inhibitor initiation more precisely by adding “days or weeks after ACS”. And we also explained it “The timing of SGLT2 inhibitor initiation varied across included studies. Most initiated therapy during the index hospitalization, typically within 3–7 days after the ACS event, while others started treatment during the early post-discharge period, up to 4 weeks after the index event. We extracted and reported these initiation windows where available to account for potential differences in treatment effect related to timing.” (2.1. Eligibility Criteria, lines 102-106).

Comments 6: Provide the text about PRISMA search results in Results section, as Figure 1 has been not mentioned in the Methods not either in the Results.

Response 6: We have now explicitly described the literature selection process in the Results section and referenced updated Figure 2 as follows: 2.2. Search Strategy and Data Sources, lines 117-118: “This systematic review and meta-analysis was conducted in accordance with the Preferred Reporting Items for Systematic Reviews and Meta-Analyses (PRISMA) 2020 guidelines and the study selection process is summarized in the PRISMA flow diagram (Figure 2).” 3.1. Study Search and Characteristics of Included Studies, lines 203-207: “The database search yielded 743 records, and 15 trials were identified through clinical trial registries. After removing duplicates and screening titles and abstracts, 475 full-text articles were assessed for eligibility and 23 studies were included in the systematic review. Finally, 16 studies met the inclusion criteria. A PRISMA 2020 flow diagram (Figure 2) illustrates the study selection process.”

Comments 7: Add a brief explanation about characteristics of participants in Summary of included studies, including but not limited to design of study, the country of study, sample number in general, age range, and percentage of studies in each level of risk of bias.

Response 7: We have now included a short descriptive summary at the beginning of the “Summary of Included Studies” section. 3.1. Study Search and Characteristics of Included Studies, lines 210-229.

“The 16 included studies comprised 5 RCTs and 11 observational studies. The characteristics of the included studies are summarized in Table 2 and Table 3. Among the observational studies, the most frequent study designs were retrospective cohorts. The included studies were conducted across diverse geographic regions, including Eu-rope, North America, and Asia, thereby reflecting diverse healthcare settings and patient populations. In total, more than 130,000 patients with ACS were analyzed, with individual study sample sizes ranging from 109 to 89,554 participants. The mean age of participants ranged from 58 to 69 years, and the majority of study populations consisted predominantly of male patients (approximately 65-80%). Regarding diabetes status, 11 studies included exclusively patients with T2DM [9-19], 1 study excluded patients with diabetes [20], and 4 studies enrolled mixed cohorts [21-24]. Most studies enrolled patients with STEMI, while NSTEMI patients were included in a smaller pro-portion. Two studies with zero events in both arms were excluded from the effect size estimation [25-26]. Risk of bias assessment showed that all RCTs were at low risk (31,25%), while among observational studies, 10 (62,5%) were rated as moderate risk, and 1 (6,25%) as low-to-moderate. However, most observational studies used statistical adjustment methods such as PSM or IPTW, and reported adjusted HRs to minimize confounding. These differences highlight the current state of research and the complexity of the available evidence, underscoring the variability and limitations of existing data regarding the effects of SGLT2 inhibitors after ACS.”

Quality of English Language

We thank the reviewer for the positive feedback regarding the language quality of the manuscript.

Additional clarifications:

Regarding the major comment: Because the aim of this study is to see the SGLT2 inhibitor effect after ACS events in patients with and without diabetes, it is critical to clearly state that in patients without diabetes, the pooled HRs of all-cause mortality, cardiovascular mortality, recurrent MI (despite its small number study observing recurrent MI as an outcome), and stroke after SGLT2 inhibitor therapy were not statistically significant. Although authors have explained it in the Conclusions, but authors must properly state these findings and provide the possible reasons for why those insignificant results were found in participants without diabetes in Discussion as well.

Response: We thank the reviewer for highlighting this point. In response, we have revised the Results and Discussion sections to more explicitly report and interpret these findings. Specifically, we specified that in the non-diabetic subgroup, the pooled hazard ratios were not statistically significant.

„In non-diabetic patients, no statistically significant effects were observed for either outcome.” Discussion, lines 454-456

We have also added a detailed discussion on the potential reasons for these observations, such as: the limited number of studies and smaller sample sizes in the non-diabetic subgroup, leading to reduced statistical power; differences in baseline cardiovascular risk and competing risk factors between diabetic and non-diabetic populations; potential variations in the pleiotropic cardiovascular effects of SGLT2 inhibitors, which may be more pronounced in the presence of metabolic dysregulation; and the possibility that longer follow-up duration may be required to detect significant benefit in lower-risk, non-diabetic cohorts. Discussion, 4.2. Limitations, lines 437-453.

               „Several factors may explain these findings. First, the number of studies and participants without diabetes was considerably smaller, limiting statistical power to detect meaningful differences. Second, the absolute CV risk is generally lower in non-diabetic populations, which may attenuate the magnitude of benefit from SGLT2 inhibition. Third, the pleiotropic cardioprotective effects of SGLT2 inhibitors (natriuresis, blood pressure reduction, and improved endothelial function) might require longer follow-ups durations or occur predominantly in patients with metabolic disturbances such as insulin resistance or hyperglycemia. Lastly, heterogeneity in baseline characteristics, timing of therapy initiation, and study design may have further diluted potential treatment effects in this subgroup.” Discussion, lines 462-471

We believe that these revisions have strengthened the discussion and improved the clinical interpretability of our findings.

Reviewer 2 Report

Comments and Suggestions for Authors

The inclusion of both randomized controlled trials and observational studies resulted in substantial heterogeneity in the analyzed data, which may negatively impact the conclusions.
Subgroup analyses by diabetes status were not performed for all endpoints, limiting understanding of the differential effect of SGLT2 inhibitors between diabetics and non-diabetics, particularly for secondary outcomes (eg, stroke, revascularization).
The effects found in the non-diabetic subgroups did not reach statistical significance and were accompanied by wide variability, but the text emphasizes the potential benefit without sufficient clarity, which is a controversial interpretation.
In the Discussion section, a number of results obtained solely from observational studies are generalized to the entire population, although for some outcomes (e.g., recurrent infarction, stroke), an effect was found only in diabetics, and no effect was confirmed in the non-diabetic population.
The study notes publication bias for the outcome of all-cause mortality (p=0.047 according to Egger's test), but the impact of this bias and methods for addressing it are not discussed in detail, despite this reducing the reliability of the pooled estimates.
The conclusion claims a reduction in mortality and MI for the entire population after ACS; however, according to their analysis, the pooled data for non-diabetics does not yield a statistically significant effect; this requires a more critical analysis and careful formulation of conclusions.
Inclusion and exclusion criteria are often given only in summary form, without disclosing how multicenter or mixed designs were handled, and the timeframes for initiation of SGLT2 inhibitor therapy in different studies are not detailed.
The lack of effect in non-diabetics is interpreted with a strong emphasis on “potential benefit”, although the confidence intervals are very wide and heterogeneity precludes firm conclusions at this stage; great caution is needed and recommendations for data generation in future RCTs are recommended.
The final statement about the advisability of early administration of SGLT2 inhibitors in all patients after ACS is not fully justified for the population without diabetes - this thesis requires significantly more qualification.
Thus, the review is based on a significant body of data, but requires improved transparency and detail in the description of methods, and greater caution in formulating general conclusions, especially for non-diabetics and for outcomes where the effect is controversial or lacks statistical significance.

Author Response

Thank you so much for taking the time to review this manuscript. Please find the detailed responses below and the corresponding revisions highlighted in the re-submitted files.

Comments 1: The inclusion of both randomized controlled trials and observational studies resulted in substantial heterogeneity in the analyzed data, which may negatively impact the conclusions. Subgroup analyses by diabetes status were not performed for all endpoints, limiting understanding of the differential effect of SGLT2 inhibitors between diabetics and non-diabetics, particularly for secondary outcomes (eg, stroke, revascularization). The effects found in the non-diabetic subgroups did not reach statistical significance and were accompanied by wide variability, but the text emphasizes the potential benefit without sufficient clarity, which is a controversial interpretation. In the Discussion section, a number of results obtained solely from observational studies are generalized to the entire population, although for some outcomes (e.g., recurrent infarction, stroke), an effect was found only in diabetics, and no effect was confirmed in the non-diabetic population. The study notes publication bias for the outcome of all-cause mortality (p=0.047 according to Egger's test), but the impact of this bias and methods for addressing it are not discussed in detail, despite this reducing the reliability of the pooled estimates. The conclusion claims a reduction in mortality and MI for the entire population after ACS; however, according to their analysis, the pooled data for non-diabetics does not yield a statistically significant effect; this requires a more critical analysis and careful formulation of conclusions. Inclusion and exclusion criteria are often given only in summary form, without disclosing how multicenter or mixed designs were handled, and the timeframes for initiation of SGLT2 inhibitor therapy in different studies are not detailed. The lack of effect in non-diabetics is interpreted with a strong emphasis on “potential benefit”, although the confidence intervals are very wide and heterogeneity precludes firm conclusions at this stage; great caution is needed and recommendations for data generation in future RCTs are recommended. The final statement about the advisability of early administration of SGLT2 inhibitors in all patients after ACS is not fully justified for the population without diabetes - this thesis requires significantly more qualification. Thus, the review is based on a significant body of data, but requires improved transparency and detail in the description of methods, and greater caution in formulating general conclusions, especially for non-diabetics and for outcomes where the effect is controversial or lacks statistical significance.

Response 1: Thank you for comments. We have revised the manuscript and added the following clarifications:

  1. Heterogeneity due to mixed study designs and overgeneralization from observational data:

We acknowledge that combining randomized controlled trials and observational studies introduced heterogeneity. Our rationale was to provide a comprehensive synthesis of the currently available evidence, given the limited number of dedicated RCTs. To address this concern, we have reported heterogeneity for each meta-analysis and performed subgroup analyses by study type, which demonstrated that results were similar, even though effect sizes varied. We have now added a dedicated paragraph in the Discussion section to address this limitation and explain how it may influence the interpretation of pooled results. Also, revised the section to distinguish which findings are driven primarily by observational data and which are supported by RCT evidence. We have revised sections of the Discussion to clearly distinguish which findings are driven primarily by observational data and which are supported by RCT evidence. (Discussion, 4.2. Limitations, lines 501-526).

“The inclusion of RCTs and observational studies broadened the evidence base and enhanced external validity, but it also increased variability in patient selection, treatment strategies, and outcome definitions. This heterogeneity may have attenuated the precision of pooled estimates and should be considered when interpreting our findings. Moreover, several outcomes, including recurrent MI and stroke, were primarily driven by observational evidence, which may be subject to residual confounding and therefore should not be generalized to the entire ACS population without confirmation from randomized trials. Due to the population included, subgroup analyses by diabetes status could not be performed for certain secondary outcomes, including rehospitalization for ACS and revascularization rates, as for these outcomes the population was represented only by diabetic patients. As a result, the differential effect of SGLT2 inhibitors in these outcomes remains unclear. Although trends toward benefit were observed in non-diabetic populations, these results did not reach statistical significance and were characterized by substantial heterogeneity and wide confidence intervals. As such, any potential benefit in this subgroup should be interpreted with great caution and regarded as a hypothesis rather than a conclusion.”

  1. Subgroup analyses by diabetes status not available for all endpoints:

We have stated in the 4.2. Limitations section that subgroup analyses by diabetes status could not be performed for certain secondary outcomes, limiting the depth of interpretation for these endpoints. However, for these outcomes, all studies included only patients with diabetes, therefore subgroup analysis could not be performed. 4.2. Limitations, lines 517-521

“Due to the population included, subgroup analyses by diabetes status could not be performed for certain secondary outcomes, including rehospitalization for ACS and revascularization rates, as for these outcomes the population was represented only by diabetic patients. As a result, the differential effect of SGLT2 inhibitors in these outcomes remains unclear.”

We revised the Discussion section to explicitly emphasize that the effects observed in non-diabetic patients were not statistically significant and were characterized by wide confidence intervals and substantial heterogeneity. Where significant effects were limited to diabetic patients, we now explicitly state this and avoid extrapolating results to the entire population.

We have replaced the text with more cautious interpretations. Discussion, lines: 452-471.

„This meta-analysis, including both RCTs and observational studies, shows that, in the overall population, SGLT2 inhibitors initiation after ACS is associated with reduction in all-cause and CV mortality. In non-diabetic patients, no statistically significant effects were observed for either outcome. In patients with T2DM, the reduction remained significant for all-cause mortality but not for CV mortality. Regarding recurrent MI, a benefit was identified among patients with T2DM, but not for the overall cohort. These findings suggest a potential role in post-ACS secondary prevention in diabetic populations. Early initiation, particularly during the acute or subacute post-ACS phase, may mitigate ischemia-reperfusion injury and enhance myocardial recovery.

Several factors may explain these findings. First, the number of studies and participants without diabetes was considerably smaller, limiting statistical power to detect meaningful differences. Second, the absolute CV risk is generally lower in non-diabetic populations, which may attenuate the magnitude of benefit from SGLT2 inhibition. Third, the pleiotropic cardioprotective effects of SGLT2 inhibitors (natriuresis, blood pressure reduction, and improved endothelial function) might require longer follow-ups durations or occur predominantly in patients with metabolic disturbances such as insulin resistance or hyperglycemia. Lastly, heterogeneity in baseline characteristics, timing of therapy initiation, and study design may have further diluted potential treatment effects in this subgroup.”

  1. Publication bias and its implications:

We have expanded the section on publication bias, discussing its potential impact on the reliability of pooled estimates and noting that it may lead to overestimation of treatment effects for all-cause mortality, and therefore the benefit should be interpreted with caution.

 3.2.4. Publication Bias Analysis, lines 324-342

“For all-cause mortality, Egger’s regression test was performed to assess publication bias and showed possible small-study effects (t=2.29 p = 0.047). Although this p-value is marginally below the conventional threshold (p < 0.05), the result does not strongly support the presence of funnel plot asymmetry. To further evaluate the robustness of our findings, we visually inspected the funnel plots, assessed the certainty of evidence using the Grades of Recommendation, As-sessment, Development and Evaluation (GRADE) evidence profile (Table 4), and conducted a leave-one-out sensitivity analysis. These approaches highlight the need for cautious interpretation of the observed mortality benefit, as it may be influenced by selective reporting. To quantify the potential impact, we applied the trim-and-fill method. The adjusted pooled HR for all-cause mortality was HR = 0.69 (95% CI: 0.58–0.83), slightly lower than the original estimate HR = (0.77 [0.67–0.89]). This finding indicates that publication bias may have led to a modest overestimation of the effect size. However, the direction of the effect remained unchanged, supporting the robustness of the observed association be-tween SGLT2 inhibitor therapy and reduced all-cause mortality. In contrast, for CV mortality, Egger’s regression test showed no significant small-study effects (p=0.66). Future large-scale RCTs and the publication of negative or neutral re-sults are essential to provide a more balanced and accurate estimate of the effect of SGLT2 inhibitors on all-cause mortality after ACS.”

  1. Reformulation of conclusions:

The Conclusion has been substantially revised to provide a more nuanced and cautious summary of the evidence, particularly for non-diabetic patients and for outcomes without significant effects. Conclusions, lines 537-555.

„SGLT2 inhibitors are associated with a reduction in all-cause and CV mortality, as well as recurrent MI, following ACS, particularly among patients with T2DM. The benefits in all-cause mortality and recurrent MI were consistent and clinically meaningful in patients with T2DM, whereas similar trends observed in non-diabetic population, where the evidence remains limited and inconclusive, with pooled estimates not reaching statistical significance and characterized by substantial heterogeneity. No significant effects were observed for stroke, revascularization, or rehospitalization for ACS. Differences between RCTs and observational studies further highlight the influence of study design on effect estimates and underscore the need for cautious interpretation. These findings suggest that the observed mortality reduction may be largely driven by patients with T2DM, underscoring the potential role of glycemic status in modulating the therapeutic response to SGLT2 inhibition. Although SGLT2 inhibitors were associated with reduced CV mortality in the overall population, the results did not achieve statistical significance, indicating a possible class effect that predominantly manifests in diabetic patients. While initiation of therapy post-ACS appears beneficial in diabetic patients, current evidence does not support universal early use in non-diabetic individuals. Large, dedicated RCTs, especially in non-diabetic post-ACS populations, are essential to clarify the risk–benefit profile, confirm these findings, and define the role of SGLT2 inhibitors in future post-ACS management strategies.”

  1. Inclusion criteria and therapy initiation timing:

We have improved the Methods section to clarify inclusion/exclusion criteria in more detail, explain how designs were handled, and provide additional information on the timing of SGLT2i initiation across studies. 2.1. Eligibility Criteria, lines 87-113

„Inclusion criteria: We included RCTs and observational studies with a follow-up ≥ 6 months. The population of interest was represented by patients (≥ 18 years) admitted for ACS, including (UA, STEMI or NSTEMI). Eligible studies evaluated treatment with SGLT2 inhibitors versus control (placebo or standard medical treatment) after the index ACS event and were included if they reported at least one of the outcomes. The primary outcomes were all-cause mortality and CV mortality following ACS in patients treated with SGLT2 inhibitors (death from any cause and death due to cardio-vascular causes). Secondary outcomes included: recurrent myocardial infarction (non-fatal MI after the index hospitalization), rehospitalization for ACS, revascularization procedures and stroke incidence. For synthesis, studies were grouped according to study design (RCT vs. observational), population characteristics (T2DM, mixed cohorts or no T2DM) and timing of SGLT2 inhibitor initiation after the index ACS event (during hospitalization vs. early post-discharge). Subgroup analyses were performed by study design (RCT vs. observational) and based on diabetes status (T2DM vs. non-diabetic). In studies with mixed population, we included only the ACS subgroup is stratified results were available. 

The timing of SGLT2 inhibitor initiation varied across included studies. Most initiated therapy during the index hospitalization, typically within 3–7 days after the ACS event, while others started treatment during the early post-discharge period, up to 4 weeks after the index event. We extracted and reported these initiation windows where available to account for potential differences in treatment effect related to timing.

Exclusion criteria: We excluded studies enrolling patients without a documented ACS event, studies lacking comparator group, articles with no publicly available full-text data for extraction, meta-analyses, case-reports, studies with unavailable outcome data, animal studies and studies evaluating SGLT2 inhibitors on patients with stable coronary artery disease (CAD), as well as, those in which outcomes could not be extracted separately. If subgroup-specific outcomes were not reported, such studies were excluded from the meta-analysis.”

  1. Recommendations for future research:

We have strengthened the Discussion to emphasize the need for well-designed, adequately powered RCTs in non-diabetic populations and for specific secondary outcomes. 4.3. Implications for Future Research, lines 528-535

„Strict adherence to SGLT2 inhibitors reduces cardiovascular hospitalizations and mortality in T2DM patients with heart failure, ACS, or those undergoing revascularization or cardiac surgery [44]. Given the substantial residual cardiovascular risk in patients with T2DM, current findings support SGLT2 inhibitors as part of secondary prevention strategies in T2DM after ACS. However, evidence in non-diabetic patients remains inconclusive. Ongoing and future RCTs will be essential to clarify their role in non-diabetic populations, to define their place in post-ACS management algorithms and guide clinical decision-making.”

Quality of English Language

We thank the reviewer for the positive feedback regarding the language quality of the manuscript.

Round 2

Reviewer 1 Report

Comments and Suggestions for Authors

The authors have answered all of my comments properly. 

My minor comments are:

  1. Modify the phrase "diabetic patients" to "patients with diabetes".
  2. Modify the phrase "non-diabetic patients" to "patients without diabetes".
  3. Provide the information about what were PSM or IPTW (Line 226).

Author Response

Comments 1: The authors have answered all of my comments properly. My minor comments are:

  1. Modify the phrase "diabetic patients" to "patients with diabetes".
  2. Modify the phrase "non-diabetic patients" to "patients without diabetes".
  3. Provide the information about what were PSM or IPTW (Line 226).

Response: We thank the reviewer for the helpful suggestions. We have replaced the term "diabetic patients" with "patients with diabetes" and "non-diabetic patients" with "patients without diabetes" throughout the manuscript. We have also added a brief explanation of PSM (propensity score matching) and IPTW (inverse probability of treatment weighting) in Line 226 of the revised manuscript, as requested.